# Post-acute sequelae of COVID-19 in residents in long-term care homes: Examining symptoms and recovery over time

Gordana Rajlic[1]*, Janice M. Sorensen[1], Benajir Shams[1], Armin Mardani[1], Ketki Merchant[1], Akber Mithani[1,2]

1 Long-Term Care and Assisted Living, Fraser Health Authority, Surrey, British Columbia, Canada,
2 Department of Psychiatry, University of British Columbia, Vancouver, British Columbia, Canada

* gordana.rajlic@fraserhealth.ca

## Abstract

### Background

Post-COVID-19 condition (PCC) has been studied extensively since the inception of the COVID-19 pandemic. In the population of long-term care (LTC) home residents, however, information about PCC and recovery after the acute phase of COVID-19 is lacking. This study contributes evidence about symptoms over time in 459 residents in nine Canadian LTC homes.

### Methods

In a comprehensive retrospective chart review, we recorded medical symptoms in a 4-week period before contracting COVID-19 ("PRE-COVID") and during 24 weeks after contracting infection (a 4-week "ACUTE-COVID" period and five subsequent 4-week periods "POST1–5"). We investigated the number and type of symptoms over time, examined different "recovery trajectories", and compared the characteristics of residents across different trajectories.

### Results

In the sample overall, the number of different symptoms increased from PRE-COVID to ACUTE-COVID (mean difference of 3 symptoms, $p<.001$), returning to the PRE-COVID level within the first two months post-infection. An individual-level examination revealed that after ACUTE-COVID about a quarter of residents did not return to their symptom baseline. There was no statistically significant difference in demographic characteristics or PRE-COVID comorbidities across different recovery trajectories. Comparing the group of residents that did not return to their symptom baseline and the group that did, the risk for not returning to baseline increased with the number of symptoms in ACUTE-COVID (adjusted for age, sex, and PRE-COVID comorbidities,

**Data availability statement:** Data used in the study was collected by Fraser Health for health care purposes and are restricted by the Freedom of Information and Protection of Privacy Act (FIPPA) regulations. Data cannot be shared publicly because there are ethical and privacy restrictions on sharing a de-identified data set (i.e. the data contains potentially identifying and sensitive information). Researchers who meet the criteria for access to confidential data are encouraged to contact the Fraser Health Department of Evaluation and Research Services (research.approvals@fraserhealth. ca) to discuss how their data request can be facilitated.

**Funding:** The study received funding from the COVID-19 Pandemic Response and Impact Grant (Co-RIG) Program of the Foundation for Advancing Family Medicine and the Canadian Medical Association Foundation, Grant/Award Number: FAFM-2021-0094-EN_Mithani. (A.M. and J.S.) https://fafm.cfpc.ca/corig/ There was no additional internal or external funding received for this study. The funders had no role in study design, data collection and analysis, decision to publish, or preparation of the manuscript.

**Competing interests:** The authors have declared that no competing interests exist.

exp[$B$]=1.15, 95% CI [1.05;1.25], $p$=.002). Additionally, there was a greater increase in the number of symptoms from PRE-COVID to ACUTE-COVID in the former group (significant interaction effect, $p$<.001). We present symptom types in each time-period.

## Conclusions

Group-level results indicated that the number of symptoms after contracting COVID-19 fell to the pre-COVID level within the first two months post-infection. An examination of individual-level symptom trajectories contributed a more granular picture of recovery after infection and characteristics of residents across different trajectories.

---

## Introduction

Since the start of the COVID-19 pandemic, a substantial body of literature has been published about post-COVID health conditions. This research was initiated by observations that some individuals who survive the acute phase of the infection experience persistence or progression of one or more symptoms or develop new symptoms beyond the acute illness phase, and do not return to baseline levels of health functioning [1–4]. The condition, named post-COVID-19 condition (PCC), post-acute sequelae of SARS-CoV-2 infection, or long COVID [5], has been studied in different populations. However, in the population of long-term care (LTC) home residents, evidence about PCC is lacking [6–9]. LTC residents have been disproportionally affected by the COVID-19 pandemic, with a large number of deaths recorded in LTC homes during the pandemic [10–13]. While post-infection mortality declined over years, concerns about functioning and health outcomes among residents who survived the COVID-19 infection were raised. Valuable overall information about recovery after COVID-19 has been provided, but information about individual recovery trajectories, heterogeneity in recovery, post COVID-19 symptoms, and PCC in the LTC population is scarce [14,15].

Furthermore, in the general population, where PCC has been studied extensively, many unknowns remain in relation to risks, etiology, manifestations, and prevalence of PCC [3,16–18]. A lack of a widely accepted definition in the past impacted research outcomes and comparability of research findings [4]. Consequently, obtaining robust PCC evidence, greatly needed to inform practice, has been challenging. In LTC residents, there are additional challenges impeding PCC investigation [7,19] – in this population, multimorbidity, frailty, and deterioration in physical and cognitive functioning is common irrespective of COVID infection, and health fluctuations are frequent. Therefore, research approaches that focus on individual trajectories and incorporate more detailed information from both pre- and post-COVID-19 time are particularly relevant in this population. The current study was conducted to provide more evidence for an improved understanding of PCC in residents in LTC homes and help in identifying, assessing, managing, and alleviating the condition.

## Methods

### The study

Following a retrospective longitudinal design, we conducted a comprehensive chart review and recorded information about clinical symptoms before and after COVID-19 infection in LTC residents who survived the acute phase of the infection. The goals of the study were to: 1) describe the symptoms in COVID-19 survivors over 24 weeks after contracting infection, 2) explore different post-acute symptom trajectories, with the focus on residents who did not return to their pre-COVID symptom baseline, and 3) compare symptoms and characteristics in residents following different recovery trajectories.

### Participants

The participants in the study were 459 residents who tested positive for COVID-19 (had a positive result on any SARS-CoV-2 microbiological test) during a COVID-19 outbreak in nine LTC homes in British Columbia, Canada, in outbreaks spanning from the start of the pandemic to the end of 2021 (March 01, 2020 – December 30, 2021). Only residents who survived the acute COVID-19 infection were included in the study, specifically, residents that survived at least two months after contracting COVID-19 (this period was sufficient for determining if residents survived the most critical phase of the disease and met the study inclusion criteria). Demographic and co-morbidity characteristics of the residents are summarized in Table 1. The LTC homes participating in the study are affiliated with Fraser Health Authority, and only LTC homes were included (i.e., nursing homes, providing 24-hour professional service to meet residents' complex care needs) while assisted living facilities were not included in the study.

### Procedure and outcome measures

For each resident, we collected information from seven 4-week periods over 28 weeks: a 4-week period before contracting COVID-19 ("PRE-COVID"), a 4-week period after the positive test (named tentatively "ACUTE-COVID"), and subsequent five 4-week periods of post-acute follow-up ("POST1" to "POST5"), as presented in Fig 1a. Overall, starting from the day of a positive COVID-19 test, information was collected over 24 weeks. Breaking the study period into multiple shorter time periods aided in assessing symptom fluctuations/stability over time. In all study periods, we collected information about clinical symptoms (the type and number of symptoms) as recorded in the residents' medical charts that include medical and care progress notes. Symptoms from various domains of functioning, such as respiratory function; nutrition/gastrointestinal; psychological and cognitive functioning; skin; and general symptoms (e.g., pain, fever, chills, fatigue) were recorded (for a comprehensive list see S1 Text). The number of symptoms in each study period was the sum of the different symptoms in the given period (how many different types of symptoms were recorded in the medical chart).

For the PRE-COVID period, information about symptoms (i.e., COVID-19 unrelated symptoms) was collected before the date of a positive COVID-19 test or the date that onset of COVID-19 symptoms was reported, whichever was earlier, as noted in the chart. For PRE-COVID, we also collected information about diagnosed comorbidities, including metabolic, cardiovascular, pulmonary, musculoskeletal, gastrointestinal, and neurological diseases (classified following the Resident Assessment Instrument – Minimum Data Set 2.0, RAI-MDS 2.0); demographic characteristics (i.e., sex and age); and activities of daily living functioning (ADL scale score, as recorded in the resident's last RAI-MDS 2.0 assessment before contracting COVID-19).

**Symptom Trajectories.** In addition to examining characteristics and outcome in all participants (at the group level), our goal was to explore individual post-acute recovery trajectories. We were interested in exploring how many residents did not return to their baseline symptoms level after ACUTE-COVID and the characteristics of these residents. Specifically, we examined the trajectory of "*not* returning to baseline" after ACUTE-COVID – labeled as trajectory T1, and the trajectory

**Table 1. Demographic, functional, and clinical characteristics at baseline.**

| Characteristics | Overall (n = 459) | T1 (n = 109) | T2 (n = 207) | T3 (n = 107) | Stat. test, p-value |
|---|---|---|---|---|---|
| **Sex**, n (%) | | | | | |
| Female | 259 (56.9%) | 64 (58.7%) | 116 (57.1%) | 55 (51.4%) | |
| Male | 195 (42.9%) | 45 (41.3%) | 86 (42.4%) | 52 (48.6%) | |
| Other | 1 (0.2%) | 0 | 1 (0.5%) | 0 | $\chi^2$=2.47, p=.65 |
| **Age**[a], M (SD) | 84.5 (10.6) | 84.3 (11.4) | 84.8 (9.8) | 82.9 (11.3) | F=1.13, p=.32 |
| **Age Category**[b], n (%) | | | | | |
| <60 yrs | 12 (2.6%) | 4 (3.7%) | 3 (1.5%) | 4 (3.7%) | |
| 61-70 years | 29 (6.4%) | 7 (6.4%) | 12 (5.9%) | 9 (8.4%) | |
| 71-80 years | 87 (19.1%) | 19 (17.4%) | 40 (19.7%) | 23 (21.5%) | |
| 81-90 years | 178 (39.1%) | 45 (41.3%) | 81 (39.9%) | 41 (38.3%) | |
| 91-100 years | 143 (31.4%) | 33 (30.3%) | 63 (31%) | 30 (28%) | |
| >100 years | 6 (1.3%) | 1 (0.9%) | 4 (2.0%) | 0 | $\chi^2$=5.8, p=.83 |
| **ADL Score**[c], M (SD) | 15.7 (8.1) | 14.21 (8.1) | 14.37 (8.4) | 13.3 (7.9) | F=0.54, p=.58 |
| **Comorbidities**, n (%) | | | | | |
| Endocrine/Metabolic | 195 (42.5%) | 52 (47.7%) | 85 (41.1%) | 43 (40.2%) | $\chi^2$=1.62, p=.45 |
| Cardiovascular | 324 (70.6%) | 74 (67.9%) | 150 (72.5%) | 70 (65.4%) | $\chi^2$=1.83, p=.40 |
| Musculoskeletal | 241 (52.5%) | 55 (50.5%) | 109 (52.7%) | 55 (51.4%) | $\chi^2$=1.46, p=.93 |
| Neurological | 397 (86.5%) | 98 (89.9%) | 173 (83.6%) | 96 (89.7%) | $\chi^2$=3.56, p=.17 |
| Psychological/Mood | 157 (34.2) | 41 (37.6%) | 74 (35.7%) | 29 (27.1%) | $\chi^2$=3.18, p=.20 |
| Pulmonary | 85 (18.5%) | 21 (19.3%) | 34 (16.4) | 21 (19.6) | $\chi^2$=0.66, p=.72 |
| Liver/Kidney | 88 (19.2%) | 26 (23.9%) | 36 (17.4%) | 14 (13.1%) | $\chi^2$=4.34, p=.11 |
| Gastrointestinal | 124 (27%) | 30 (27.5%) | 50 (24.2%) | 32 (29.9%) | $\chi^2$=1.28, p =.53 |
| Cancer | 62 (13.5%) | 20 (18.3%) | 23 (11.1%) | 10 (9.3%) | $\chi^2$=4.74, p=.09 |
| **# of Comorbidities**, M (SD) | 3.6 (1.5) | 3.8 (1.7) | 3.5 (1.4) | 3.5 (1.4) | F=1.9, p=.16 |
| **# of Comorbidities ≥ 3**, n (%) | 353 (76.9%) | 84 (77.1%) | 154 (74.4%) | 81 (75.7%) | $\chi^2$=0.28, p=.87 |

Note: T1 – "not returning to baseline" trajectory; T2 – "returning to baseline" trajectory; T3 – "other" trajectory (residents that could not be classified in T1 or T2); n = 423 for the three trajectories (symptoms information missing or outcome unknown for 36 residents). For Sex and Age, missing n = 4.

a, b Age at the date of positive COVID-19 test.

c ADL score = Activities of Daily Living Scale – Long Form score, from the RAI-MDS 2.0 (within 6 months before follow-up start date, the score was available for 383 residents) – the score ranges from 0 to 28, and a higher score indicates lower ADL function.

of "returning to baseline" after ACUTE-COVID – labeled as trajectory T2. We used the following procedure: Based on information about symptoms in different time periods for each resident, we compared the number of symptoms in each of the post-acute periods (POST1 – POST5) to the number of symptoms in PRE-COVID. We recorded if there was an increase in number of symptoms in the POST1 – POST5 periods as compared to PRE-COVID, and in how many periods there was any increase in symptoms. Based on this information, we classified the residents into trajectories T1 or T2, as depicted in Fig 1b and 1c.

Residents that did not fit into the T1 or T2 classification as described, were classified separately and labeled as trajectory T3 ("other"). This included residents with symptoms fluctuating from period to period (i.e., increase in the number of symptoms in some periods and return to baseline in other periods), and with an unclear pattern of recovery/deterioration over follow-up. Our focus in this research was on learning about residents who clearly did not return to baseline symptoms after ACUTE-COVID (T1 group), and on comparing them to all other residents. However, it also was deemed useful to compare T1 to T2 (return to baseline trajectory); therefore, further differentiation among residents (i.e., T2 and T3) was needed.

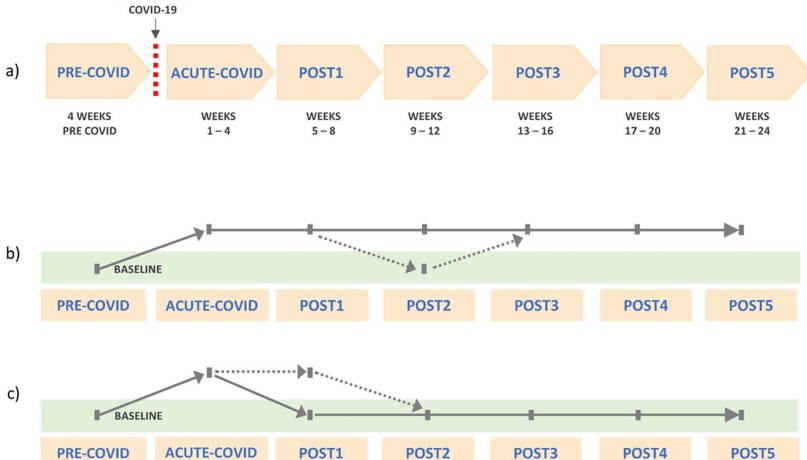

**Fig 1. Study design. (a)** Time periods in the study in which symptoms were recorded. **(b)** Trajectory T1. *Not returning to baseline* trajectory, defined as "greater number of symptoms in each period after ACUTE-COVID in comparison to PRE-COVID level (baseline), or returning to baseline in only one (any) out of the five POST periods". **(c)** Trajectory T2. *Returning to baseline* trajectory, defined as "not exhibiting a greater number of symptoms as compared to baseline in any POST period, or returning to baseline within the first 3 months after contracting COVID (the latest in POST2)".

## Data collection

Residents' administrative and medical records (electronic medical records or paper charts in the LTC homes) were accessed by three research assistants (RA) with a clinical background. RAs completed training on standard protocols for data collection, they reviewed training materials, completed preliminary chart review, and discussed the results of preliminary data collection. Following the training, information was collected for ten individual records, to assess interrater reliability [20] (interclass correlation coefficient was calculated, with the criterion of ICC ≥.90 used for satisfactory reliability). After a satisfactory reliability was established, RAs proceeded with sample data collection [21]. All medical and care notes in residents' charts with information relevant for the study were reviewed. Information about COVID-19 outbreaks and COVID-19 testing data were obtained from Population and Public Health in Fraser Health and participating LTC homes.

## Ethical considerations

The study was approved by the British Columbia Harmonized Ethics Review process (UBC REB Number H21-02350) prior to starting. In the data collection process, information was accessed that could identify individual participants (i.e., the current study is a chart review study, and identifiable information was essential to the research). The data were anonymized upon collection and all appropriate measures were taken to protect the privacy of individuals and to safeguard the information. Medical records were accessed and reviewed for relevant information in the period from March 2022 to January 2023.

In the current study, the need for informed consent to participate has been waived by the Research Ethics Committee. Conditions were present that required departures from general principles of consent, as outlined in the Tri-Council Policy Statement: Ethical Conduct for Research Involving Humans – TCPS 2 [22], which sets the standard for ethics in Canadian research. Specifically, our study was a retrospective review of medical records in a large sample of residents in long-term care homes. In this unique population, characterized by older age, adverse medical conditions, and short life expectancy/high mortality, informed consent to participate would be impossible or impracticable to obtain for a large number of residents. That is, many residents died before the study initiation (they fulfilled the study eligibility criteria – survived the acute phase of COVID-19 and had the required follow-up information, but deceased before study initiation). Many residents

were not able to provide informed consent due to their medical conditions (e.g., dementia). Information resulting from this research, on the other hand, was important for understanding resident needs and informing post-acute COVID-19 care. Research did not involve more than minimal risk to the participants and the lack of the participant's consent was unlikely to adversely affect the welfare of the participant. Based on the study context and the reasons that include those listed above, a waiver of consent to participate was requested in the research ethics application, and it was approved by the Research Ethics Committee.

### Data analysis

We described resident characteristics and comorbidities PRE-COVID, and symptoms in each time-period, with descriptive statistics and visual summaries provided. We compared proportions of residents with and without clinical symptoms across periods by nonparametric test for change over time (McNemar test). The mean number of symptoms was compared over time by conducting repeated measures ANOVA, with targeted pairwise comparisons reported (LSD post hoc test). After classifying residents into different recovery trajectories, we summarized characteristics of residents following those trajectories and compared the groups by conducting chi-square test or one-way ANOVA, depending on the nature of variable in question (Bonferroni correction was used when multiple comparisons were conducted, such as in comorbidities comparisons). We compared the number of symptoms in different groups over follow-up and reported between-subjects effect in ANOVA (group-differences overall and at the different time points). Finally, several variables relevant for post-COVID-19 recovery (e.g., risk factors for T1 vs T2 trajectory) were examined in multivariate logistic regression analysis, with adjusted coefficients reported. Outlying data points and residuals from the analyses were examined to ensure the appropriateness of the conducted analysis. Missing data details are included in S1 Text.

## Results

### Baseline demographic, functional, and clinical characteristics

At the time of the positive COVID test, median age of the residents was 86.5 years, with most residents in the age category of 81–90 years, followed by the age category of 91–100 years (Table 1). We provide demographic characteristics, ADL score, and comorbidities pre-COVID in Table 1. The most prevalent comorbidities pre-COVID were neurological and cardiovascular disorders. Almost all residents (92.8%) had more than one comorbidity. Specific types of comorbidities are presented in Fig 2, with hypertensive disease (57.2% of residents) and dementia (48.6%) indicated as the most common.

### Symptoms over time

A substantial proportion of residents (67.2%) experienced clinical symptoms in the PRE-COVID period. There was a statistically significant increase in the number of residents experiencing any clinical symptoms from PRE-COVID to ACUTE-COVID, with most residents (93.8%) experiencing at least one symptom in ACUTE-COVID ($\chi^2$ = 94.7, $p$ <.001). Following ACUTE-COVID, the number of residents with clinical symptoms gradually decreased towards PRE-COVID numbers. In POST1, there was still significantly more residents exhibiting clinical symptoms than in PRE-COVID (75.3% in POST2, $\chi^2$ = 8.96, $p$ =.003), whereas there was no difference between PRE-COVID and any of POST2 – POST5 (all $p$-values >.05).

Regarding number of symptoms, the mean number of different symptoms experienced by residents in all time periods is presented in Fig 3 (orange colour line, labeled "overall"). There was a statistically significant change in the number of symptoms over time (omnibus test for change over time $F$ = 187.29, $p$ <.001, $\eta^2$ =.32). In comparison to PRE-COVID, there was a substantially greater number of symptoms in ACUTE-COVID (mean difference 2.97, $SE$ =.14, $p$ <.001), and there was a small increase from PRE-COVID to POST1 (mean difference of 0.27, $SE$ =.11, $p$ =.02). The number of symptoms in POST2 – POST5 was similar to the number of symptoms in PRE-COVID, with no statistically significant differences recorded (all $p$-values >.05).

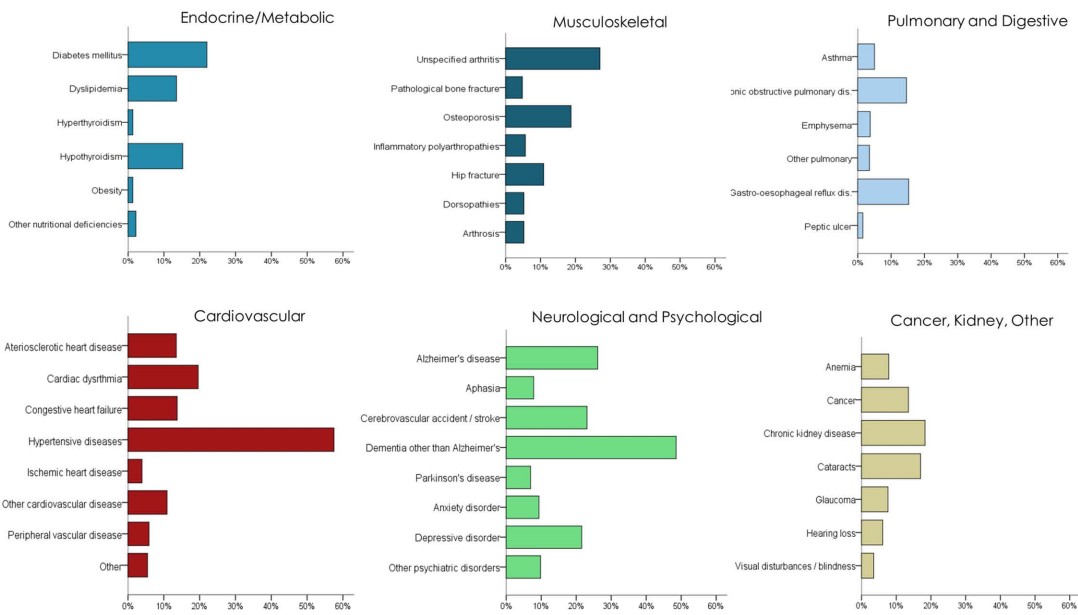

**Fig 2. Specific comorbidities at baseline.** Percentage of LTC residents diagnosed with specific diseases is presented (n = 459).

In terms of type of symptoms during the ACUTE-COVID period, the most prevalent symptom among residents was cough (recorded in 69.8% of all residents), followed by low food intake (54% of residents), chest congestion (33.5%), and fatigue (30%). Presence of different types of symptoms in ACUTE-COVID and in all study periods is shown in Fig 4.

### Characteristics of residents following different post-acute trajectories

Based on comparing the number of symptoms in post-acute periods (POST1-POST5) to the number of symptoms in PRE-COVID (as described in the Methods), 109 residents (25.8%) were classified as T1 ("*not* returning to baseline" trajectory, depicted in Fig 1b) and 207 residents (48.9%) were classified as T2 ("returning to baseline" trajectory, depicted in Fig 1c). Remaining residents were classified as T3 ("other" group, *n* = 107, 25.3%). Characteristics of residents in the T1 trajectory, along with T2 and T3, are presented in Table 1. The groups were similar in terms of sex, age, and PRE-COVID ADL score (see Table 1). There were no statistically significant differences found between T1 and the other two groups in types of comorbidities present in PRE-COVID and in the mean number of comorbidities (Table 1). Similarly, regarding specific comorbidities (Fig 2), the proportion of residents in T1 presenting with the most common comorbidities – hypertensive disease (59.6%), dementia (52.3%), and diabetes (28.4%), was similar to those in the other two groups (all *p*-values >.05). Time when COVID-19 outbreak was declared in the respective LTC home and the resident contracted infection (i.e., year 2020 vs 2021) was not related to the trajectory type ($\chi^2$ =.08, *p* =.96).

The number of symptoms in different trajectories over time is presented in Fig 3. There was a statistically significant relation between the trajectory type and the number of symptoms over time (trajectory main effect: *F* = 39.4, *p* <.001, $\eta^2$ =.17). Between-group comparisons in POST periods indicated that the T1 group experienced more symptoms in all periods (POST1-POST5) as compared to T2 (all *p*-values <.001) and T3 (all *p*-values <.01 except in POST5, when *p* =.06). In ACUTE-COVID, T1 had significantly more symptoms than T2 (*M* = 5.3, *SD* = 3.1 and *M* = 4.3, *SD* = 2.6 respectively, *p* =.002) while the difference with T3 was marginally significant (*p* =.07).

In the multivariate logistic analysis that included age, sex, number of different comorbidities in PRE-COVID, and number of symptoms in ACUTE-COVID, the only statistically significant predictor of T1 vs T2 was number of symptoms

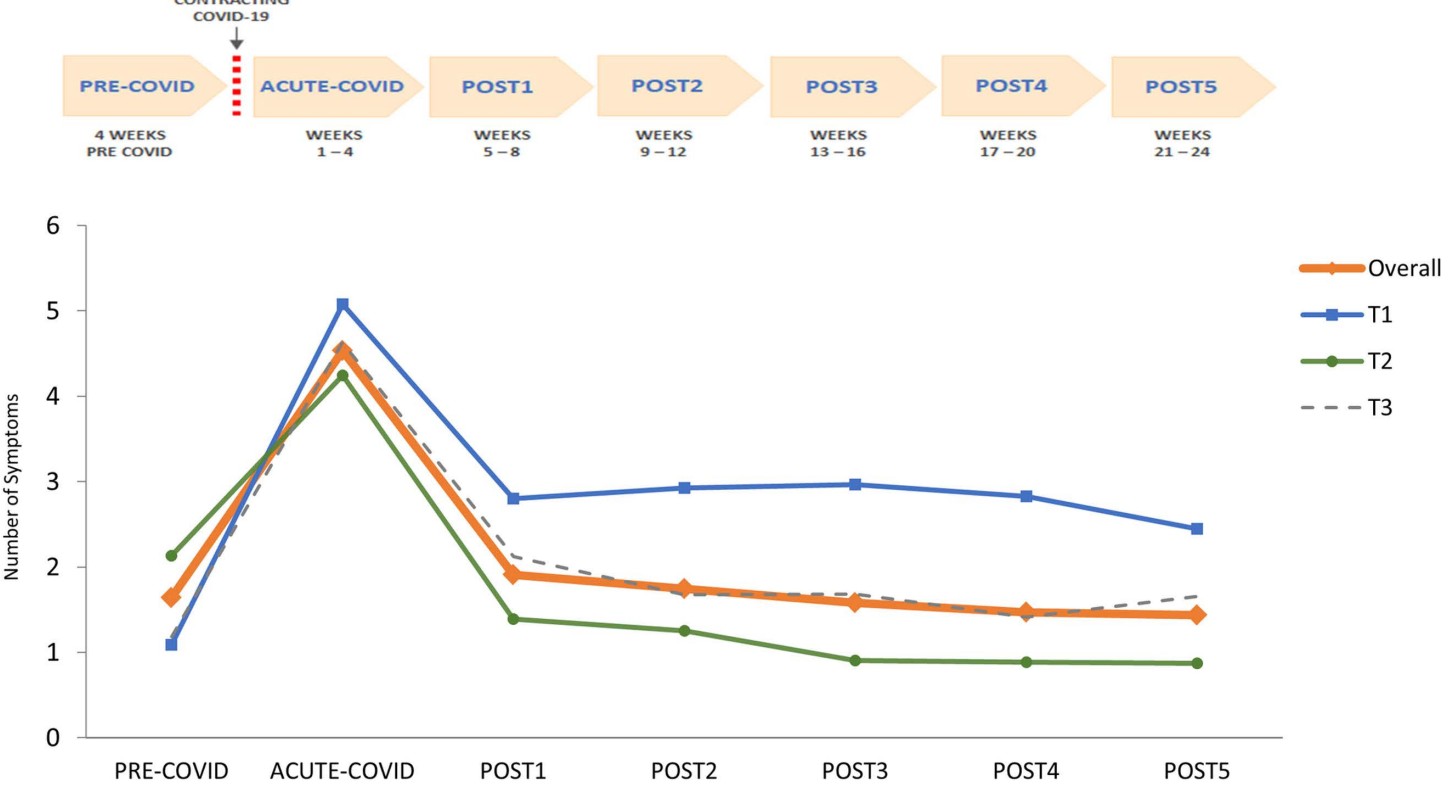

**Fig 3. Number of symptoms over study periods.** The number of different symptoms over the time periods is presented for all residents with information about symptoms available (orange line, *n* = 433), along with the number of symptoms for residents classified into different post-acute recovery trajectories, as defined in Procedure – blue line for T1 trajectory (*n* = 109), green line for T2 (*n* = 207), and grey line for T3 (*n* = 107).

in ACUTE-COVID; adjusted *b* =0.14, *p* =.002, exp(*B*) = 1.15 (95% CI 1.05–1.25). In a supplementary analysis conducted, a statistically significant interaction effect was found between the trajectory type and the change in number of symptoms from PRE-COVID to ACUTE-COVID (*F* = 34.6, *p* <.001), with a larger change from PRE-COVID to ACUTE-COVID recorded in T1 (Fig 3).

A review of symptom types revealed that, similarly to the sample overall, the most frequently experienced symptoms in ACUTE-COVID in the T1 group were cough, low food intake, chest congestion, and fatigue (S1 Fig). Over subsequent POST periods, the most frequent symptoms were low food intake and agitation, the same two symptoms that were the most frequent in the sample overall (Fig 4). Other symptoms persistent over multiple POST periods (present in a smaller number of T1 residents) were rashes, edema, pressure ulcers, and pain.

## Discussion

Numerous factors are associated with post-COVID-19 recovery. In the LTC home residents population, characterized by complex conditions and care needs, challenges in investigation of PCC are pronounced. While addressing some of these challenges, in the current study, we examined medical symptoms over time in 459 LTC residents who survived COVID-19 infection. We examined symptoms over six months after contracting COVID-19 infection and in a period before infection, while focusing on short time increments (i.e., a 4-week PRE-COVID period, a 4-week ACUTE-COVID period, and five 4-week periods subsequent to ACUTE-COVID, see Fig 1). Group-level results indicated that the number of symptoms substantially increased from PRE-COVID to ACUTE-COVID and gradually declined to the PRE-COVID levels within

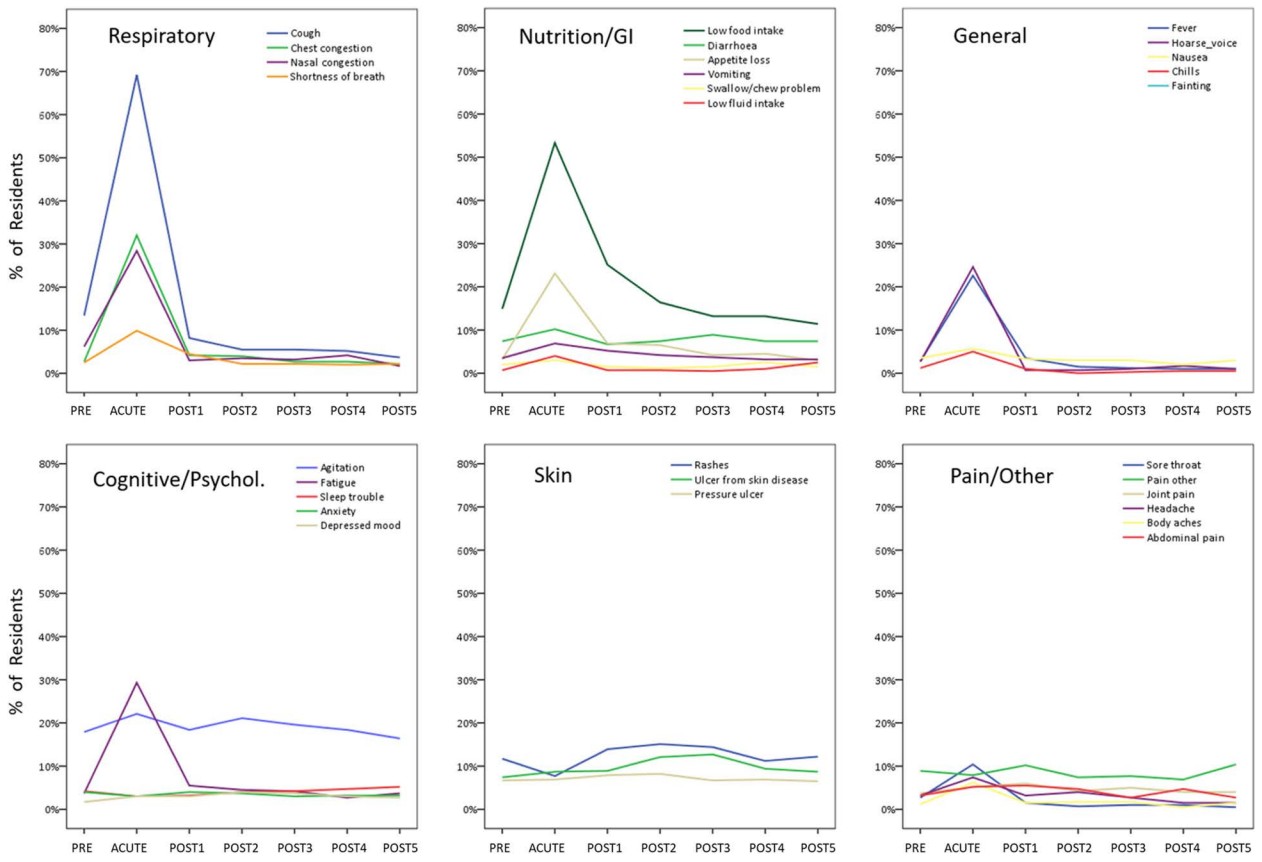

**Fig 4. Types of symptoms over study period.** In the overall sample, percentage of residents experiencing the symptoms in each time-period is presented (valid *n* = 433).

the first two months after contracting COVID-19 infection. Over the rest of the follow-up period, there was no significant increase in number of symptoms as compared to PRE-COVID. These findings, in the overall sample, are consistent with previous findings about post-COVID-19 outcomes, which pointed to an increase in clinical care needs in LTC residents with a history of COVID-19 in the initial period after infection, followed by outcomes similar to those in residents with no history of COVID-19 [14,15]. A closer look at individual recovery trajectories in the current study, however, added valuable details to this general picture.

The results revealed that even though many COVID-19 survivors (about a half of them) returned to their PRE-COVID symptom level after ACUTE-COVID, a quarter of the residents did not return to their baseline symptom level over the length of the follow-up period (labeled T1 group). In between-group comparisons, the T1 group had a higher number of symptoms in all post-acute time increments compared to the other residents. This group could meet some criteria proposed in the general definition of PCC [4], and the size of this group is comparable to some PPC prevalence estimates in the general population [4,23], albeit with great variability in estimates in the literature so far [3,24]. Regarding symptom type, the most frequent symptoms found over follow-up in the T1 (i.e., low food intake and agitation) were also predominant in the group of residents that did return to baseline. This may suggest that, in LTC residents, PCC could be seen as an acceleration of the process of health decline that is already present to a certain degree and characterizes this population rather than a unique emerging condition. Additional symptoms noted (e.g., rashes, peripheral edema) are consistent with post-infective processes with multisystem involvement, as PCC was described in previous research [25,26].

There were no significant differences between the T1 group and other residents in pre-COVID demographic characteristics and comorbidities. What distinguished T1 residents from those who returned to their PRE-COVID baseline symptom levels were a significantly greater number of symptoms in ACUTE-COVID and a greater change in the number of symptoms from PRE-COVID to ACUTE-COVID. This suggests that this group of residents possibly had a more severe COVID-19 disease; to confirm this, additional evidence is needed, such as about treatment and hospitalizations after contracting infection [27]. It should be noted that this research was conducted with LTC residents who survived the acute phase of COVID-19 (i.e., at least two months after infection); therefore, these residents were more resilient and able to better deal with an infection that proved devastating for many in this population. The results of the current study suggest that more serious infections, even in more resilient residents, may still be related to more serious outcomes. These results further suggest that adequate interventions and reducing severity in the acute phase of COVID-19 remain a priority for improving post-COVID outcomes in this population. In the general population, greater number of symptoms in the acute-COVID-19 phase was also indicated as a significant predictor of PCC [28].

In relation to post COVID-19 care, in this study, the most common symptoms after ACUTE-COVID were general symptoms of health decline, found in geriatric populations after a wider range of adverse health events, including COVID-19 [29–31]. General rehabilitation programs may be beneficial in post-COVID care of LTC residents, as suggested in recent research [8,32]. Finally, it should be underscored that median age in our sample was 85 years. Multimorbidity was widespread and most residents exhibited some clinical symptoms before contracting COVID-19. In terms of generalizability and comparisons with PCC in the general adult population, the unique characteristics of the LTC home residents must be thoughtfully considered.

This study was limited in scope and not all factors that could be relevant for post-COVID outcomes were examined, e.g., treatment and care received during and after acute illness and vaccination uptake. Vaccination uptake was very high in LTC homes in the study region once COVID-19 vaccines were available [33]. The effects of these factors should be examined in future research. More detailed examination of residents with unclear recovery trajectories (i.e., T3 group in this study) is needed; in this study, we focused on residents that clearly did not return to their PRE-COVID symptom baseline (T1 group). We utilized a retrospective longitudinal chart review study design, and prospective design with standardized recording of symptoms (and on a finer scale) could provide further evidence about PCC outcomes. Additionally, for a full picture of PCC, information directly reported by the residents should be examined along with chart review. Finally, it should be underscored that in this study we focused only on residents with a history of COVID-19. Due to chart review process requirements, reviewing symptoms over time for residents without a history of COVID-19 was not feasible at this time. Therefore, we do not know for how many residents without a history of COVID-19 the number of symptoms increases over a comparable time period for reasons other than COVID-19 – such information should be collected in future research and would aid interpretation of the results of the current study.

## Conclusion

In conclusion, the current study provided evidence about post-COVID-19 symptoms and recovery in LTC residents. We found that in the study participants overall, the average number of symptoms after infection returned to the PRE-COVID level within the first two months post-infection. The group of residents that did not return to their PRE-COVID symptom baseline over follow-up, compared to the group that did, had a larger number of symptoms in ACUTE-COVID, and there was a greater change in the number of symptoms from PRE-COVID to ACUTE-COVID in this group. The most frequent symptoms in ACUTE-COVID were cough, low food intake, chest congestion, and fatigue, while the most frequent over follow-up were low food intake and agitation. Details about post-COVID-19 clinical symptoms and characteristics of residents facing greater challenges in their post-COVID-19 recovery contribute to the overall picture of post-COVID-19 outcomes in LTC residents. Such evidence is needed in LTC clinical and care practice and is beneficial to LTC residents and their families. It helps informing care planning and ensuring appropriate support for residents' quality of life and comfort after COVID-19 infection.

## Supporting information

**S1 Text. Additional information.**
(PDF)

**S1 Fig. Symptoms in T1 trajectory.**
(PDF)

## Acknowledgments

The authors acknowledge Adriaan Windt, Clayon Hamilton, Ian Cameron, Jennifer Walls, and academic partners Shannon Freeman, Simon Carroll, Karen Davison, Maura MacPhee, and Valorie Crooks for their contributions to the study initiation/conceptualization. We appreciate the support of Ronald Kelly, Simran Dhadda, Bassem Daniel, and Andrea Lai with data retrieval and data collection, and Ian Fyffe and Sherin Jamal with proofreading and reviewing. We thank Fraser Health's Population and Public Health for their help in obtaining relevant data, and we would like to express our great appreciation to the participating long-term care homes for their support in providing data access. The views and conclusions expressed in this paper do not necessarily reflect those of the Fraser Health Authority.

## Author contributions

**Conceptualization:** Gordana Rajlic, Janice M. Sorensen, Akber Mithani.

**Data curation:** Gordana Rajlic, Benajir Shams, Armin Mardani, Ketki Merchant.

**Formal analysis:** Gordana Rajlic.

**Funding acquisition:** Janice M. Sorensen, Akber Mithani.

**Investigation:** Gordana Rajlic, Janice M. Sorensen, Benajir Shams, Armin Mardani, Ketki Merchant.

**Methodology:** Gordana Rajlic, Janice M. Sorensen, Akber Mithani.

**Project administration:** Janice M. Sorensen, Benajir Shams, Akber Mithani.

**Resources:** Janice M. Sorensen, Benajir Shams, Akber Mithani.

**Software:** Gordana Rajlic, Benajir Shams, Armin Mardani, Ketki Merchant.

**Supervision:** Janice M. Sorensen, Benajir Shams, Akber Mithani.

**Validation:** Gordana Rajlic, Janice M. Sorensen, Benajir Shams, Armin Mardani, Ketki Merchant.

**Visualization:** Gordana Rajlic.

**Writing – original draft:** Gordana Rajlic, Janice M. Sorensen.

**Writing – review & editing:** Gordana Rajlic, Janice M. Sorensen, Benajir Shams, Armin Mardani, Ketki Merchant, Akber Mithani.

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
