## [Decision Letter · Decision Letter 0]

11 Feb 2025

PONE-D-25-01016Post-acute sequelae of COVID-19 in residents in long-term care homes: Examining symptoms and recovery over timePLOS ONE

Dear Dr. Rajlic,

Thank you for submitting your manuscript to PLOS ONE. After careful consideration, we feel that it has merit but does not fully meet PLOS ONE’s publication criteria as it currently stands. Therefore, we invite you to submit a revised version of the manuscript that addresses the points raised during the review process.

**ACADEMIC EDITOR: ** The manuscript contributes to the topic about post COVID-19 condition among elderly in LTC homes. This is critical area and very relevant toward ensuring quality care among this population not only for COVID-19 but for other flu-related conditions. The quality of the study is also near satisfactory. Please respond to the following comments in addition to the comments raised by the 2 reviewers below: 1. This study involves human subjects and therefore the ethical considerations should be clearly presented. Please create a subsection "Ethical Considerations" and consolidate all ethical decisions under this subsection.2. Ensure that the manuscript is presented in line with plosone guidelines Submission Guidelines | PLOS ONE  ==============================

We look forward to receiving your revised manuscript.

Kind regards,

Ibrahim Jahun, MD, MSC, PhD

Academic Editor

PLOS ONE

Journal Requirements:

“The study was supported by funding from the COVID‐19 Pandemic Response and Impact Grant (Co‐RIG) Program of the Foundation for Advancing Family Medicine and the Canadian Medical Association Foundation, Grant/Award Number: FAFM-2021-0094-EN_Mithani. (A.M. and J.S.)

https://fafm.cfpc.ca/corig/

4. In the online submission form, you indicated that [The data that support the findings of this study are available upon request from the corresponding author and with appropriate Fraser Health Authority institutional and British Columbia Harmonized Ethics Review approvals for secondary use of data. Medical records data are not publicly available due to institutional regulations and privacy or ethical restrictions.]. 

5. Please review your reference list to ensure that it is complete and correct. If you have cited papers that have been retracted, please include the rationale for doing so in the manuscript text or remove these references and replace them with relevant current references. Any changes to the reference list should be mentioned in the rebuttal letter that accompanies your revised manuscript. If you need to cite a retracted article, indicate the article’s retracted status in the References list and also include a citation and full reference for the retraction notice.

Reviewers' comments:

Reviewer's Responses to Questions

**Comments to the Author**

1. Is the manuscript technically sound, and do the data support the conclusions?

Reviewer #1: Yes

Reviewer #2: Yes

2. Has the statistical analysis been performed appropriately and rigorously? 

Reviewer #1: Yes

Reviewer #2: I Don't Know

3. Have the authors made all data underlying the findings in their manuscript fully available?

Reviewer #1: Yes

Reviewer #2: Yes

4. Is the manuscript presented in an intelligible fashion and written in standard English?

Reviewer #1: Yes

Reviewer #2: Yes

5. Review Comments to the Author

Reviewer #1: The manuscript addresses an important topic which was approached using an appropriate study design, and your results justify the conclusions stated, and the manuscript is well written.

I do have some questions and feedback:

pg5 ln100: According to your methods you indicate collecting information for 5 periods of 4weeks (=20 weeks) after infection. Please clarify.

pg5 ln107: Please specify the month/year due to discrepancies in the start of the epidemic

pg5 ln108-109: Why did you pick this criteria given your follow-up period was 24 weeks

pg6 ln112-113: I am not sure what the differences are between assisted living and LTC in BS but in Alberta there are considerable differences and providing different levels of care. Therefore, this last part of the sentence is confusing and contradictory.

pg7 ln138-140: I am confused about this criteria - if you are looking at information about pre-existing symptoms, why are you collecting symptoms from the date of COVID-19 symptoms onset, as any symptoms collected then could be attributed to COVID? Or perhaps you meant that symptoms were collected BEFORE the date of COVID symptoms were reported? Please explain

pg16 ln319: According to your time frame, you examined symptoms before, during covid and then 5 months after. Please correct.

pg16 ln333-334: But you only measured upto months after covid ?

Reviewer #2: This study contributes evidence about symptoms over time in 459 residents in nine Canadian LTC homes., and has certain study significance. However, the following concerns should be addressed:

1、This retrospective study may introduce potential biases into the collected data.

2、This study indicates that the author gathered data on clinical symptoms documented in hospital medical records. Would you like to know whether there are corresponding standards for reference for these symptom information collected in the paper? Such as cognition, anxiety, etc. Are these symptoms assessed on a scale or?

Thank you!

6. PLOS authors have the option to publish the peer review history of their article (what does this mean? ). If published, this will include your full peer review and any attached files.

**Do you want your identity to be public for this peer review?** For information about this choice, including consent withdrawal, please see our Privacy Policy .

Reviewer #1: No

Reviewer #2: No

---

## [Author Response · Author response to Decision Letter 0]

27 Feb 2025

Responses to Editor’s and Reviewers’ comments are submitted as document titled “Responses to Reviewers” (this is a shortened version here)

Journal Requirements:

1.

We reviewed the guidelines again and made some corrections in compliance with the PLOS ONE’s style requirements (changes tracked in the manuscript).

2.

We clarified that this reported funding was all funding/support received for the study. There was no additional external or internal funding received for this study.

3.

Updated Data Availability statement:

Data used in the study was collected by Fraser Health for health care purposes and are restricted by the Freedom of Information and Protection of Privacy Act (FIPPA) regulations. Data cannot be shared publicly because there are ethical and privacy restrictions on sharing a de-identified data set (i.e. the data contains potentially identifying and sensitive information). Researchers who meet the criteria for access to confidential data are encouraged to contact the Department of Evaluation and Research Services (research.approvals@fraserhealth.ca) to discuss how their data request can be facilitated.

4.

Our data can not be made publicly available for both ethical and legal reasons. These are secondary data (medial records data) released to us from Fraser Health Authority only for the purpose of conducting research. Additional approval is needed for any other uses of data. The protocol approved by the research ethics board did not include data sharing, and additional approval from the ethics is also required, upon review for privacy and ethical considerations for data sharing.

To respond to your additional inquiry, we contacted the regulating bodies and in consultation with them, we are providing an updated Data availability statement (the statement is more specific and answers your questions). If any additional information is needed, please let us know.

Updated Data Availability statement:

Data used in the study was collected by Fraser Health for health care purposes and are restricted by the Freedom of Information and Protection of Privacy Act (FIPPA) regulations. Data cannot be shared publicly because there are ethical and privacy restrictions on sharing a de-identified data set (i.e. the data contains potentially identifying and sensitive information). Researchers who meet the criteria for access to confidential data are encouraged to contact the Department of Evaluation and Research Services (research.approvals@fraserhealth.ca) to discuss how their data request can be facilitated.

5.

We checked our reference list. We did not cite retracted papers.

Reviewers' comments:

Reviewer #1:

pg5 ln100: According to your methods you indicate collecting information for 5 periods of 4weeks (=20 weeks) after infection. Please clarify.

On pg5 ln100, we stated ‘…to describe the symptoms in COVID-19 survivors over 24 weeks after infection’. We changed to ‘…over 24 weeks after contracting infection’. Thank you for pointing to this, it needed to be specified here. These 24 weeks could be further broken down to a 4-week acute-COVID and 20 weeks after the acute phase (post-acute time consisting of 5 POST periods of 4 weeks = 20 weeks). This was explained later in the text (Procedure) and here, the purpose was just to make a general statement referring to the time after contracting infection.

The total time, including PRE-COVID, ACUTE-COVID, and 5 post-acute POST periods, was 28 weeks. This was depicted visually in Fig 1, and we added reference to Fig 1 to several paragraphs, as visual presentation is helpful to distinguish these time periods.

pg5 ln107: Please specify the month/year due to discrepancies in the start of the epidemic

We added specific dates to the manuscript: from March 01, 2020 to December 31, 2021.

pg5 ln108-109: Why did you pick this criteria given your follow-up period was 24 weeks

This was our study inclusion (eligibility) criteria – we wanted to include residents who survived the acute disease (‘COVID-19 survivors’), and to examine symptoms in this group of residents only. It was deemed that 2-month period was sufficient in determining that the resident was beyond the most critical phase of the disease. For these residents (who met our eligibility criteria) we then collected information over 28 weeks time, in the defined study periods (i.e., pre-COVID, acute-COVID, post-COVID time). We expanded this section a bit to provide more detail on our inclusion criterion.

Previously, we conducted another study focusing on mortality after contracting COVID-19 (during the acute phase and over a longer period after) and we provided evidence in this respect. This current study was intended to focus on COVID-19 survivors only (and their symptoms pre, during, and after COVID), that is why we had such study eligibility criteria. If in this group mortality occurred, it was reported (in Missing data), and this mortality was not part of our study inclusion-exclusion criteria.

pg6 ln112-113: I am not sure what the differences are between assisted living and LTC in BS but in Alberta there are considerable differences and providing different levels of care. Therefore, this last part of the sentence is confusing and contradictory.

Wording that we used here was confusing - we reworded so that it is clearer that we included LTC homes only.

“The LTC homes participating in the study are affiliated with Fraser Health Authority (FHA), and only LTC homes were included (i.e., homes providing 24-hour professional service to meet residents’ complex care needs) while Assisted Living facilities were not included in the study.”

pg7 ln138-140: I am confused about this criteria - if you are looking at information about pre-existing symptoms, why are you collecting symptoms from the date of COVID-19 symptoms onset, as any symptoms collected then could be attributed to COVID? Or perhaps you meant that symptoms were collected BEFORE the date of COVID symptoms were reported? Please explain

Yes, symptoms were collected BEFORE the day any COVID-19 symptoms were reported (if COVID-19 symptoms were reported before positive COVID-19 test date). The current wording is:

“For the PRE-COVID period, information about symptoms (i.e., COVID-19 unrelated symptoms) was collected before the date of positive COVID-19 test or the date that onset of COVID-19 symptoms was reported, whichever of was earlier, as noted in the chart”.

pg16 ln319: According to your time frame, you examined symptoms before, during covid and then 5 months after. Please correct.

Yes, that is correct. We added ‘see Figure 1’ at the end of this sentence. In Figure 1, time periods are visually presented, therefore referring to the figure should be helpful. We also added ‘contracting’ into that sentence:

“We examined symptoms over six months after contracting COVID-19 infection and in a period before infection, while focusing on short time increments (i.e., a 4-week PRE-COVID period, a 4-week ACUTE-COVID period, and five 4-week periods subsequent to ACUTE-COVID, see Fig 1).”

pg16 ln333-334: But you only measured upto 5 months after covid?

We reworded slightly, the specific time reference was removed here – it was not necessary, as specifics of the time frame were explained in many places throughout the manuscript.

“The results revealed that even though many COVID-19 survivors (about a half of them) returned to their PRE-COVID symptom level after ACUTE-COVID, a quarter of the residents did not return to their baseline symptom level over the length of the follow-up period”.

To Reviewer 1: Thank you very much for thoroughly reading through the manuscripts and for your comments. They were helpful to us, to see where clarification was needed and to improve the manuscript.

Reviewer #2:

1、This retrospective study may introduce potential biases into the collected data.

The current study was designed as retrospective examination of the information in residents’ medical charts. While prospective design is preferred, this was not realistic for us in the context of the current project. Rather, we decided to use information that was already there, in residents’ charts, collected from the onset of the pandemic. Information from regular medical examinations is valuable, and we utilized it to learn more about recovery after surviving the acute phase of COVID-19.

We were aware and observant of the risk for bias in the given research design, and we tried to minimize it by following our research protocol and procedure. In limitations, in Discussion section, we stated:

“We utilized a retrospective longitudinal chart review study design, and prospective design with standardized recording of symptoms could provide further evidence about PCC outcomes. Additionally, for a full picture of PCC, information directly reported by the residents should be examined along with chart review”

We hope that more information and in particular prospective research will be available about this topic in near future.

2、This study indicates that the author gathered data on clinical symptoms documented in hospital medical records. Would you like to know whether there are corresponding standards for reference for these symptom information collected in the paper? Such as cognition, anxiety, etc. Are these symptoms assessed on a scale or?

As you noted, we collected information about clinical symptoms as documents in residents’ medical records. Detailed information, such as are scale-assessed symptoms is not typically present in medical records. Therefore, we collected data about the presence/absence of the symptoms, as it was indicated in the charts. A composite score (the number of different symptoms) was the main measure that we used in the study.

It should be noted that symptoms were recorded by LTC professionals, according to their professional standards. For the current retrospective examination, we just collected this information, without any further interpretation/modification.

Upon reviewer’s comment, we added to our limitations paragraph (Discussion), and stated:

“We utilized a retrospective longitudinal chart review study design, and prospective design with standardized recording of symptoms, and on a finer scale, could provide further evidence about PCC outcomes.”

To conclude, we agree that future research that utilizes a finer scale as opposed to dichotomous categorization will provide more information about this important topic.

Thank you for your thoughtful comments.

While revising your submission, please upload your figure files to the Preflight Analysis and Conversion Engine (PACE) digital diagnostic tool, https://pacev2.apexcovantage.com/.

We completed this.

---

## [Decision Letter · Decision Letter 1]

4 Mar 2025

Post-acute sequelae of COVID-19 in residents in long-term care homes: Examining symptoms and recovery over time

PONE-D-25-01016R1

Dear Dr. Rajlic,

We’re pleased to inform you that your manuscript has been judged scientifically suitable for publication and will be formally accepted for publication once it meets all outstanding technical requirements.

Kind regards,

Ibrahim Jahun, MD, MSC, PhD

Academic Editor

PLOS ONE

Additional Editor Comments (optional):

Reviewers' comments:

Reviewer's Responses to Questions

**Comments to the Author**

1. If the authors have adequately addressed your comments raised in a previous round of review and you feel that this manuscript is now acceptable for publication, you may indicate that here to bypass the “Comments to the Author” section, enter your conflict of interest statement in the “Confidential to Editor” section, and submit your "Accept" recommendation.

Reviewer #1: All comments have been addressed

Reviewer #2: All comments have been addressed

2. Is the manuscript technically sound, and do the data support the conclusions?

Reviewer #1: Yes

Reviewer #2: Yes

3. Has the statistical analysis been performed appropriately and rigorously? 

Reviewer #1: Yes

Reviewer #2: I Don't Know

4. Have the authors made all data underlying the findings in their manuscript fully available?

Reviewer #1: Yes

Reviewer #2: (No Response)

5. Is the manuscript presented in an intelligible fashion and written in standard English?

Reviewer #1: Yes

Reviewer #2: Yes

6. Review Comments to the Author

Reviewer #1: Thank you for your responses, which have adequately addressed all of the reviewer feedback. Thank you for this research, which addresses an underserved but worthy population.

Reviewer #2: The author has completed the revision. I have no other questions. Thank you!

7. PLOS authors have the option to publish the peer review history of their article (what does this mean? ). If published, this will include your full peer review and any attached files.

**Do you want your identity to be public for this peer review?** For information about this choice, including consent withdrawal, please see our Privacy Policy .

Reviewer #1: No

Reviewer #2: No

---

## [Editor Report · Acceptance letter]

PONE-D-25-01016R1

PLOS ONE

Dear Dr. Rajlic,

I'm pleased to inform you that your manuscript has been deemed suitable for publication in PLOS ONE. Congratulations! Your manuscript is now being handed over to our production team.

Kind regards,

on behalf of

Dr. Ibrahim Jahun

Academic Editor

PLOS ONE